 SHORT REPORT

# Light-induced engagement of microglia to focally remodel synapses in the adult brain

Carla Cangalaya[1,2,3], Stoyan Stoyanov[3], Klaus-Dieter Fischer[2], Alexander Dityatev[3,4,5]*

[1]ESF International Graduate School on Analysis, Imaging and Modelling of Neuronal and Inflammatory Processes, Magdeburg, Germany; [2]Institut für Biochemie und Zellbiologie, Otto-von-Guericke-University, Medical Faculty, Magdeburg, Germany; [3]Molecular Neuroplasticity, German Center for Neurodegenerative Diseases (DZNE), Magdeburg, Germany; [4]Medical Faculty, Otto-von-Guericke University, Magdeburg, Germany; [5]Center for Behavioral Brain Sciences (CBBS), Magdeburg, Germany

**Abstract** Microglia continuously monitor synapses, but active synaptic remodeling by microglia in mature healthy brains is rarely directly observed. We performed targeted photoablation of single synapses in mature transgenic mice expressing fluorescent labels in neurons and microglia. The photodamage focally increased the duration of microglia-neuron contacts, and dramatically exacerbated both the turnover of dendritic spines and presynaptic boutons as well as the generation of new filopodia originating from spine heads or boutons. The results of microglia depletion confirmed that elevated spine turnover and the generation of presynaptic filopodia are microglia-dependent processes.

*For correspondence:
alexander.dityatev@dzne.de

**Competing interests:** The authors declare that no competing interests exist.

## Introduction

Microglia are highly motile cells that regularly survey their environment within the central nervous system and are responsible for immune defence in the brain (*Nimmerjahn et al., 2005*). Activation of microglia in response to infection or during ischemia, stroke and CNS trauma results in a release of different inflammatory signaling mediators (*Davies et al., 2019*; *Greenhalgh et al., 2020*) and can be associated with synaptic elimination by microglia (*Brown and Neher, 2014*; *Wake et al., 2009*).

Also, non-activated microglia in the healthy mouse brain can interact with synapses (*Davalos et al., 2005*; *Wake et al., 2009*) and thus play a critical role in neural circuit remodeling and brain plasticity during the development of the brain (*Akiyoshi et al., 2018*; *Miyamoto et al., 2016*; *Paolicelli et al., 2011*; *Weinhard et al., 2018*). At this stage, microglia employ complement-dependent mechanisms to eliminate synapses (*Schafer et al., 2012*; *Stevens et al., 2007*). In mature brains, microglia constantly make brief direct contacts with pre- and postsynaptic elements (*Akiyoshi et al., 2018*; *Nimmerjahn et al., 2005*; *Reshef et al., 2017*; *Wake et al., 2009*). The frequency of contacts is increased during synaptic plasticity at postnatal day P30 (*Pfeiffer et al., 2016*). The interaction between microglia and neurons has been demonstrated to contribute to synaptic modulation during motor learning (P30 and P60) (*Parkhurst et al., 2013*), sensory plasticity (P30) (*Tremblay et al., 2010*), forgetting of remote memories (P70–P84) (*Wang et al., 2020*), and chronic stress-related synaptic remodeling (P60-120) (*Milior et al., 2016*).

However, the direct observation of structural plasticity caused by resting microglial contact with synapses in the healthy adult brain has been hampered due to the infrequency of sporadic structural

synaptic alterations at this stage (*Grutzendler et al., 2002*; *Holtmaat et al., 2006*). Therefore, we developed an approach to evaluate the role of microglia in synaptic remodeling by transiently attracting microglial processes to synapses in a temporo-spatially controlled manner using a laser-induced focal injury. This is an acute injury model that while subtle is likely to involve a combination of injury-specific and more general mechanisms shared by homeostatic/synaptic activity-driven microglia-synapse interactions.

We used two-photon microscopy to visualize the structural dynamics of microglia and their inter-actions with dendritic spines and axonal boutons in 16 adult (3–4 months old at the beginning of imaging) *Cx3cr1*^Cre/Tomato x GFPM *Thy-1* male mice, which expressed red fluorescent protein (tdTo-mato) in all microglia cells and enhanced green fluorescent protein (EGFP) in a subset of cortical pyramidal neurons (*Figure 1A*). The retrosplenial cortex (RSC) was chosen for imaging as an area crucial for spatial navigation, memory, and history-dependent value coding (*Hattori et al., 2019*). To avoid a complex global inflammatory response of microglia (e.g. triggered by traumatic injury or infection) and maximize the contribution of local mechanisms potentially shared by synaptic repair and experience-dependent plasticity, we employed photodamage (PD) of single synaptic elements. Three types of experiments were performed, by targeting (i) a single dendritic spine (Spine-PD) to attract microglia to the damaged, imaged dendritic site; (ii) an area near a spine (Near-Spine-PD) to promote microglial approach but avoid direct damage to the imaged spine; or (iii) an area close to an axonal bouton (Near-Bouton-PD) to attract microglia but avoid direct damage to the imaged axon (*Figure 1B*). First, we collected time-lapse images every 10 min for 2 hr under basal conditions. Then, three photodamages, one of each type, were performed, and structural changes were tracked every 10 min during an additional 2 hr (*Figure 1A*). Data from each replicate were averaged per mouse before being used for statistical analysis.

The contact area between microglia and dendrites/axons, defined by colocalization of microglia and neurons, increased after all types of photodamage in comparison with the baseline. Despite the evident colocalization of microglia with spines or boutons in our experiments, the resolution of light microscopy used here does not allow us to claim that this apparent contact represents the physical contact between these both cells. The temporal profiles of fast-onset expansion in the contact area were similar in Spine-PD and Near-Spine-PD experiments and different from the slow increase in the contact area observed in the Near-Bouton-PD experiments (*Figure 1C2,D2,E2*, and *Figure 1— source data 1*). The mean microglia-neuron contact area increased during 2 hr, from 0.53% and 0.24% before photodamage to 4.39% and 3.79% after Spine-PD and Near-Spine-PD, respectively. The microglia-neuron contact area was significantly higher in Spine-PD than in Near-Spine-PD and Near-Bouton-PD (*Figure 1—source data 2*). More detailed analysis of contacts per spine and bou-ton revealed that the number of microglia-contacted synaptic sites increased after Spine-PD and Near-Spine-PD but not after Near-Bouton-PD, compared with the baseline (*Figure 1—source data 2*). Strikingly, the contact between microglial processes and spines/boutons lasted much longer after Spine-PD (60.97 ± 3.33 min), Near-Spine-PD (50.33 ± 3.22 min), or Near-Bouton-PD (70.57 ± 6.35 min) than during baseline imaging (~23.77 ± 4.9 min for spines and 21.18 ± 2.53 min for boutons) (*Figure 1—source data 2*). The prolonged duration of the contact between microglia processes and spines/boutons found after photodamage is similar to the values reported after induction of ische-mia (*Wake et al., 2009*).

All types of photodamage resulted in a higher turnover of spines and axonal boutons compared with the baseline (*Figure 1—source data 1* and *Figure 1—Videos 1–3*), when spine and bouton turnover rates were less than 0.05%. Importantly, the temporal dynamics of turnover matched those of microglia-neuron contact during the 2 hr evaluation period (*Figure 1C2,D2,E2*, and *Figure 1— source data 1*). To summarize the remodeling of spines and boutons, we calculated a mean turnover over the 2 hr of observation. It was higher after Spine-PD compared with other photodamage types (*Figure 1—source data 2*). There were no differences between the percentages of gained and lost spines after Spine-PD and Near-Spine-PD (*Figure 1C3,D3*). However, the rate of bouton formation after Near-Bouton-PD was higher than the elimination rate (*Figure 1E3*). Interestingly, we observed the formation of spine-head filopodia and new filopodia at boutons after Spine-PD/Near-Spine-PD and Near-Bouton-PD, respectively (*Figure 1—figure supplements 1* and *2* and *Figure 1—Videos 4–6*). These structures were not observed during the baseline period and have been suggested to be important for the formation of multiple spine synapses following induction of long-term potentia-tion (*Toni et al., 1999*). The filopodia occurrence rate after Near-Bouton-PD (79.55%) was

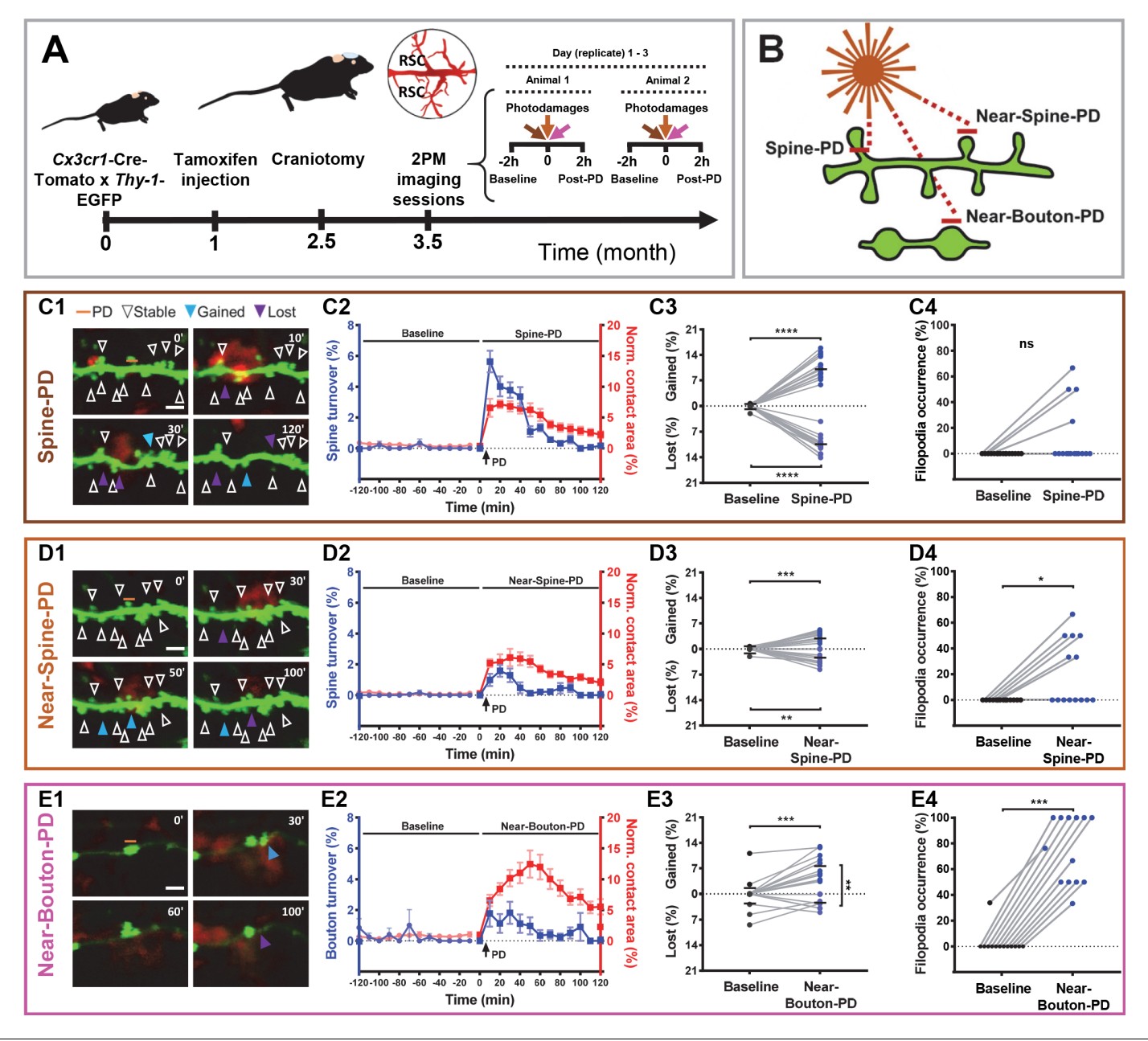

**Figure 1.** Synaptic photodamage induces microglia contact and promotes spine and bouton turnover and filopodia formation in the RSC. (**A**) Timeline of the experiments. Mice received intraperitoneal injections of tamoxifen at 1 month of age to trigger the expression of red fluorescent protein in microglia. After 1.5 months, a glass coverslip was implanted over the RSC and 1 month later imaging of the RSC was performed. Typically, the baseline (2 hr) and responses to three types of PD (2 hr) were imaged in parallel at three locations, in two animals per day. (**B**) Short horizontal bars (≈1 μm) illustrate locations of laser-irradiated areas in three types of photodamage experiments. (**C–E**) Summary graphs showing structural changes after photodamage (white line = scale bar, 3 μm). From left to right, sequential images of microglia (red) - dendrite/axons (green) interactions (yellow) before the photodamage (0 min) and 10, 30, 50, 60, and 120 min afterwards (orange line denotes position of PD laser target). Stable (white arrowheads pointing to spines/boutons that remained stable over 2 hr of imaging), gained (blue arrowheads pointing to spines/boutons appeared from time point to time point) and lost (purple arrowheads pointing to spine/boutons disappeared) structures were identified in these images (**C1,D1,E1**). Using this information, we calculated turnover rates and normalized microglia-neuron contact areas (colocalization) before and after photodamage (shown on the same plot as mean + s.e.m. with two different y-axes) (Spine-PD: n = 16 mice, 47 dendrites; Near-Spine-PD: n = 16, 35 dendrites; Near-Bouton-PD: n = 14, 34 axons). The contact area curves of baseline and after photodamage differ significantly (Spine-PD: p=0.001, Near-Spine-PD: p=0.016 and Near-Bouton-PD: p=0.040, Generalized Estimated Equation (GEE)) (**C2,D2,E2**). The colocalization area curve of Near-Bouton-PD was different in comparison with the Spine-PD and Near-Spine-PD curves (Spine-PD vs. Near-Spine-PD: p=0.687; Spine-PD vs. Near-Bouton-PD: p=0.003, and Near-

*Figure 1 continued on next page*

*Figure 1 continued*

Spine-PD vs. Near-Bouton-PD: p=0.006; GEE). The middle plot shows the average percentage change (during 2 hr) in gained and lost spines 2 hr before (baseline) and after photodamage (C3,D3,E3) (Spine-PD: n = 16 mice, 47 dendrites; Near-Spine-PD: n = 16, 35 dendrites; Near-Bouton-PD: n = 14, 34 axons). Plots on the right show the changes in filopodia occurrence (percentage of replicates per mouse where at least one filopodium was observed) after the photodamage (C4,D4,E4), paired two-sided Wilcoxon test was used for comparisons (Spine-PD, n = 16 mice, 47 dendrites; Near-Spine-PD, n = 16, 35 dendrites; Near-Bouton-PD, n = 14, 34 axons). All data points indicate the average value per mouse. Significant differences are shown as *p<0.05; **p<0.01; ***p<0.001; ns - not significant. *Figure 1—Videos 1–3* correspond to images shown in panels (C1), (D1), and (E1), respectively.

The online version of this article includes the following video, source data, and figure supplement(s) for figure 1:

**Source data 1.** Time-course analysis for microglia-neuron contact area and turnover before and after PD and between experiments (Spine-PD, Near-Spine-PD and Near-Bouton-PD).
**Source data 2.** Analysis of structural changes induced by different types of photodamage (Spine-PD, Near-Spine-PD and Near-Bouton-PD).
**Source data 3.** These data sets contain the mean values of each parameter per animal, related to each panel in *Figure 1*.
**Figure supplement 1.** Scheme illustrating filopodia quantification for spines and boutons.
**Figure supplement 2.** Time-lapse sequences of filopodia formation from spines and boutons after photodamage.
**Figure 1—video 1.** Structural changes after Spine-PD (10' - 120') compared with a basal image (0').
https://elifesciences.org/articles/58435#fig1video1
**Figure 1—video 2.** Structural changes after Near-Spine-PD (10' - 120') compared with a basal image (0').
https://elifesciences.org/articles/58435#fig1video2
**Figure 1—video 3.** Structural changes after Near-Bouton-PD (10' - 120') compared with a basal image (0').
https://elifesciences.org/articles/58435#fig1video3
**Figure 1—video 4.** Formation of transient head filopodium from a spine adjacent to the microglia-contacted structure selected for PD.
https://elifesciences.org/articles/58435#fig1video4
**Figure 1—video 5.** Head filopodia formation from the spine near to PD after microglia contact in a Near-Spine-PD experiment.
https://elifesciences.org/articles/58435#fig1video5
**Figure 1—video 6.** Formation of filopodia from both adjacent and targeted boutons during microglia contact and after Near-Bouton-PD.
https://elifesciences.org/articles/58435#fig1video6

remarkably higher than after Spine-PD (11.98%) and Near-Spine-PD (23.61%) (*Figure 1C4,D4,E4*). All the boutons which formed filopodia were apparently contacted by microglia for an average duration of 87 ± 33.7 min. Spines and boutons were classified according to their stability during the 2 hr of observation as newly formed, transient (existing spines/boutons that disappeared at least for one frame), and stable (existing spines/boutons that remained completely stable for 2 hr) (*Figure 2A1, B2,C1*). New and transient spines and boutons were closest to the photodamage site (*Figure 2A2, B2,C2* and *Figure 2A3,B3,C3*) and were more often in contact with microglia when compared with stable spines/boutons after all types of photodamage (*Figure 2A4,B4,C4*). The duration of the microglial contact was longer with the newly formed and lost spines than with the stable spines (*Figure 2A5,B5,C5*). After Near-Spine-PD the filopodia always appeared at stable spines, while after Near-Bouton-PD the filopodia originated more often at newly formed than at stable or lost boutons (*Figure 2A6,B6,C6*).

The temporal and spatial association between the photodamage-induced stable microglia-neuron contact and increased turnover of spines, boutons, and filopodia suggested that at least some of these morphological changes were mediated by microglia. To test this hypothesis, we depleted microglia in the whole brain using PLX3397 as an inhibitor of colony-stimulating factor one receptor, the activity of which is vitally important to microglial cells (*Elmore et al., 2014*; *Spiller et al., 2018*). Twelve mice were randomly allocated to two groups: six mice were fed a diet containing PLX3397 for 28 consecutive days, and the other six control animals received the same diet but without PLX3397 (*Figure 3A*). The number of microglia was dramatically reduced after 1 week of PLX3397 treatment and dropped to 0.6% after 4 weeks (*Figure 3B*, *Figure 3—figure supplement 1A–D*). During baseline imaging, we observed that PLX3397 treatment did not affect the turnover of spines/boutons, which remained very low before and after the treatment in both the control and treated groups (*Figure 3—figure supplement 2B,F,J*).

However, the depletion of microglia resulted in suppression of the spine turnover associated with photodamage. Comparing mean turnover values before and after PLX3397/control-treatment for each type of photodamage, we detected no differences in the control group (*Figure 3C2,D2,E2*), while turnover in the PLX3397-treated group decreased only after Spine-PD (19.13 ± 3.35 vs.

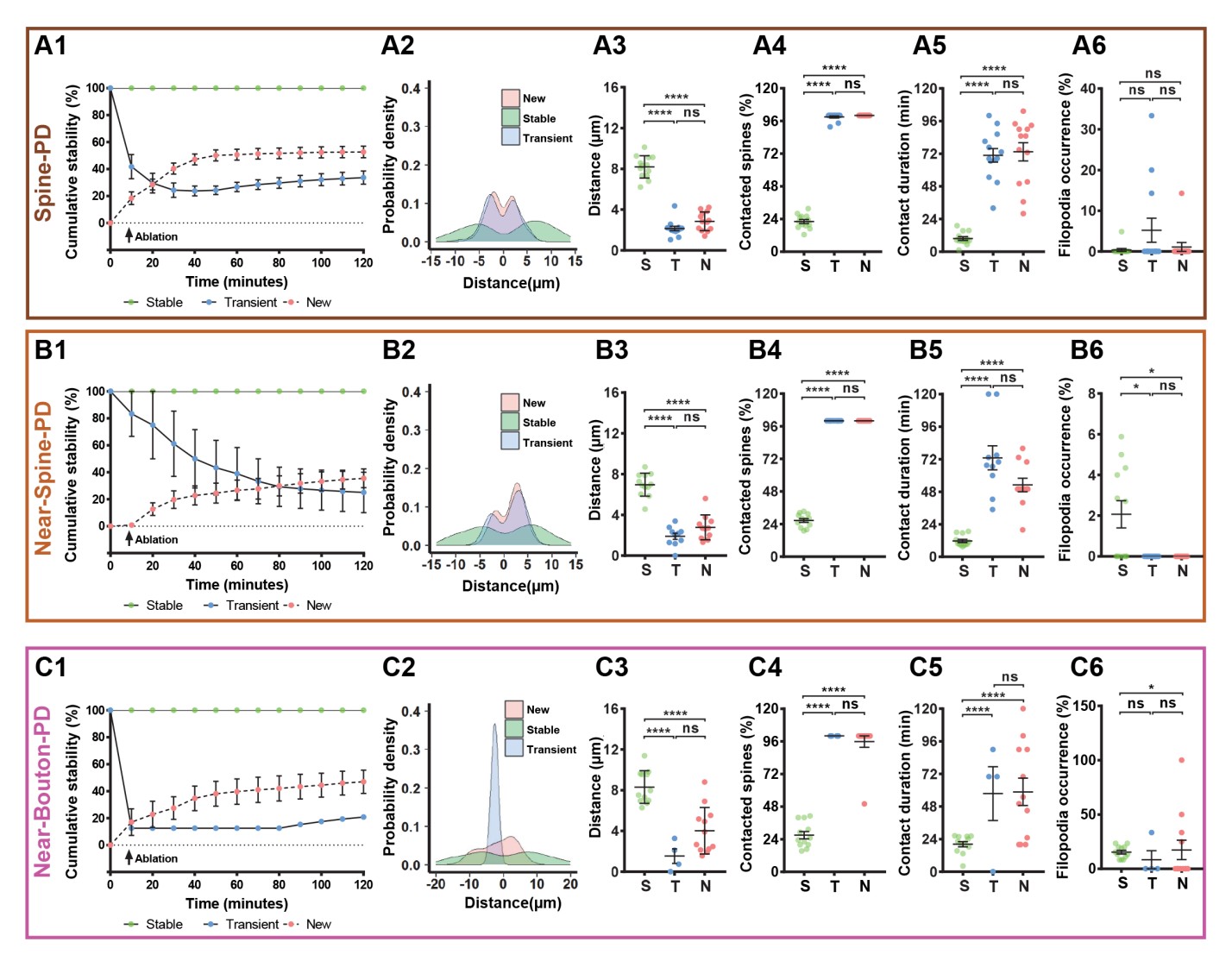

**Figure 2.** Characterization of microglia interaction with spines and boutons after photodamage. Spines or boutons (which were studied after PD of spine (**A1–A6**), near to spine (**B1–B6**) and near to bouton (**C1–C6**) were categorized according to their stability as stable (defined as the spines or boutons that persisted for 120 min), transient (defined as existing spines or boutons that can disappear or reappear during the 120 min studied) or new (defined as newly formed spines) and averaged per mouse. The total sample size was for Spine-PD: n = 16 mice, 47 dendrites, 700 spines (Stable = 506, New = 61, Transient = 133); for Near-Spine-PD: n = 16 mice, 35 dendrites, 490 spines (Stable = 439, New = 24, Transient = 27); for Near-Bouton-PD: n = 14 mice, 34 axons, 230 boutons (Stable = 196, New = 27, Transient = 7). Left graphs show cumulative stability percentage curves (**A1,B1,C1**), highlighting a separation between the three categories of spines/boutons according to their stability (dots represents the mean value per time). The distribution of spines and boutons across dendrite and axons are shown in **A2,B2,C2**. 0 value in X axis represents the position of the selected spine or bouton for photodamage, which were not included in these plots. From middle to right graphs (Spine-PD: Stable 16 mice/45 dendrites/506 spines, New 14/37/61, Transient 14/42/133; Near-Spine-PD: Stable 16 mice/35 dendrites/439 spines, New 15/20/24, Transient 13/17/27; Near-Bouton-PD: Stable 14 mice/35 axons/196 boutons, New 14/25/27, Transient 5/6/7), quantification of the distance between each individual spine/bouton with the targeted PD position (**A3,B3,C3**), the number of spines/boutons that had been contacted by microglia during 2-hour imaging session (**A4,B4,C4**), duration of microglia contact with individual spines and boutons (**A5,B5,C5**), and filopodia occurrence rate (**A6,B6,C6**) after microglial contact with damaged and near damaged spines or boutons. Means and s.e.m. as error bars are showed in all plots and each dot represents an animal. Statistical significance is represented by asterisks (*p<0.05; **p<0.01; ***p<0.001; ****p<0.0001; ns - not significant) for multiple comparisons after paired two-sided Wilcoxon test.

The online version of this article includes the following source data for figure 2:

**Source data 1.** These data sets contain the mean values of each parameter per animal, related to each panel in *Figure 2*.

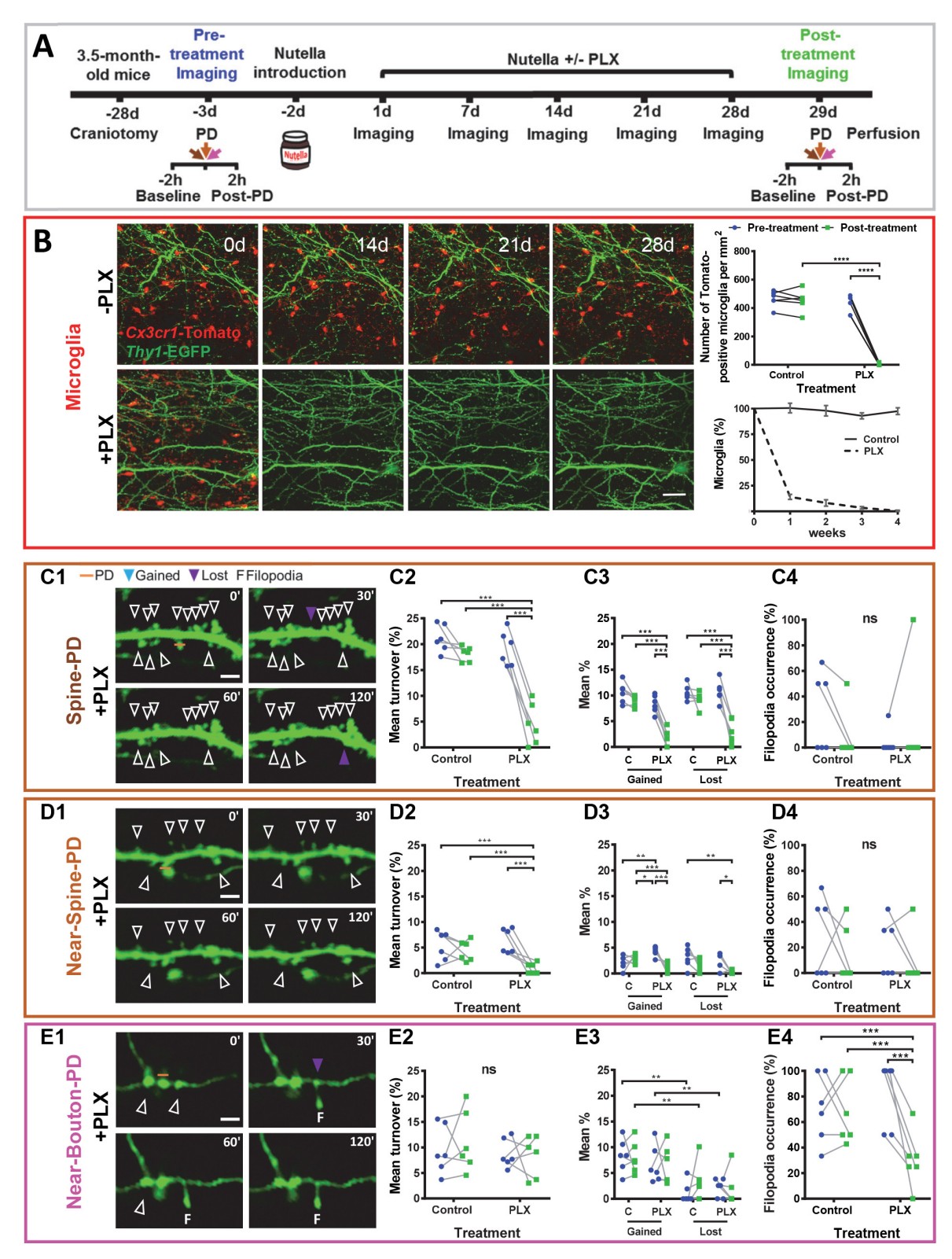

**Figure 3.** Microglia depletion prevents synaptic photodamage-induced spine turnover and bouton filopodia formation in the RSC. (**A**) Experimental design. (**B**) Representative images of microglia (red) from a control mouse (-PLX) and a mouse treated with PLX3397 (+PLX) mixed with Nutella. Scale bar, 50 µm. The upper-right graph shows the quantification of microglia depletion before (blue) and after 4 weeks of treatment with PLX3397 (green) in the control and treated groups. Significant depletion was found only after the treatment with PLX3397 (p<0.001) (n = 6 mice for each group). The lower

*Figure 3 continued on next page*

*Figure 3 continued*

graph (mean+ s.e.m.) shows the percentage of surviving microglia in control and treated mice (n = 4 mice for each group). (C,D,E) Summary results for three types of PD experiments. From left to right: representative time-lapse imaging of dendrites and axons before (0') and after the photodamage (30', 60', 120') (C1,D1,E1). PD laser targets are marked by orange lines. No evidence of microglial processes or direct colocalization was found. The following panels show the change in overall mean turnover (C2,D2,E2), mean percentage of spines gained and lost (C3,D3,E3), and filopodia occurrence (C4,D4,E4) before (blue) and after the PLX3397 treatment (green) in the control and treated groups (For Spine-PD: Control-Pre, n = 6 mice, 18 dendrites; Control-Post, n = 6, 15 dendrites; PLX-Pre, n = 6, 23 dendrites; PLX-Post, n = 6, 16 dendrites. For Near-Spine-PD: Control-Pre, n = 6 mice, 13 dendrites; Control-Post, n = 6, 18 dendrites; PLX-Pre, n = 6, 18 dendrites; PLX-Post, n = 6, 16 dendrites. For Near-Bouton-PD: Control-Pre, n = 6 mice, 16 axons; Control-Post, n = 6, 20 axons; PLX-Pre, n = 6, 14 axons; PLX-Post, n = 6, 20 axons). Data points indicate the average value per mouse. *p<0.05; **p<0.01; ***p<0.001, ns - not significant for GEE post-hoc pairwise comparison with Tukey method (*Figure 3—source datas 1* and *2*). *Figure 3—Videos 1–3* correspond to images shown in panels (C1), (D1), and (E1), respectively. See also *Figure 3—figure supplements 1* and *2*.

The online version of this article includes the following video, source data, and figure supplement(s) for figure 3:

**Source data 1.** GEE analysis of structural changes induced by different types of photodamage (Spine-PD, Near-Spine-PD and Near-Bouton-PD).
**Source data 2.** Tukey's post-hoc tests of structural changes induced by different types of photodamage (Spine-PD, Near-Spine-PD and Near-Bouton-PD).
**Source data 3.** These data sets contain the mean values of each parameter per animal, related to each panel in *Figure 3*.
**Figure supplement 1.** Ex-vivo confirmation of microglia depletion after the administration of PLX.
**Figure supplement 1—source data 1.** These data sets contain the mean value of each parameter per animal, related to each panel in *Figure 3—figure supplement 1*.
**Figure supplement 2.** Baseline evaluation and filopodia characterization.
**Figure supplement 2—source data 1.** These data sets contain the mean of each parameter per animal, related to each panel in *Figure 3—figure supplement 2*.
**Figure 3—video 1.** Structural changes after Spine-PD (10' - 120') compared with a basal image (0') in a microglia-depleted brain.
https://elifesciences.org/articles/58435#fig3video1
**Figure 3—video 2.** Structural changes after Near-Spine-PD (10' - 120') compared with a basal image (0') in a microglia-depleted brain.
https://elifesciences.org/articles/58435#fig3video2
**Figure 3—video 3.** Structural changes after Near-Bouton-PD (10' - 120') compared with a basal image (0') in a microglia-depleted.
https://elifesciences.org/articles/58435#fig3video3

4.51 ± 3.97, p<0.001, Tukey's post-hoc test) and Near-Spine-PD (6.38 ± 2.40 vs. 0.93 ± 1.06, p<0.001, Tukey's post-hoc test) and not after Near-Bouton-PD (*Figure 3C2,D2,E2*, *Figure 3— source datas 1* and *2* and *Figure 3—Videos 1–3*). Detailed analysis of the turnover revealed that depletion of microglia reduced the mean percentages of gained and lost spines after Spine-PD and Near-Spine-PD (*Figure 3C3,D3,E3*) and of bouton filopodia induced by Near-Bouton-PD (*Figure 3C4,D4,E4*). There was, however, no effect of PLX3397 on filopodia length (*Figure 3—figure supplement 2C,G,K*) or stability (*Figure 3—figure supplement 2D,H,L*). Also, spine-head filopodia formation was independent of microglia (*Figure 3C4,D4*).

These observations provide evidence that microglia react rapidly to photodamaged synapses in the mature brain and establish prominent contacts with synaptic structures to promote spine turnover, in agreement with studies of sporadic synaptic remodeling in the neocortex and somatosensory cortex of adult mice (*Akiyoshi et al., 2018*; *Parkhurst et al., 2013*; *Wake et al., 2009*). We think that our approach with targeting of single synaptic sites makes an important advance to minimize the injury and standardize the signal that acutely drives microglial approach and interaction with neurons, as compared to previous studies using single-cell or single-neurite laser-mediated photoablation (*Canty et al., 2013*; *Go et al., 2016*; *Haynes et al., 2006*; *Stoyanov et al., 2020*). An important component of that signal, as shown in single-cell photoablation studies, is photodamage-released adenosine triphosphate (ATP) that attracts microglia via activation of P2Y12 receptors (*Haynes et al., 2006*). This is also the mechanism of microglial attraction under physiological conditions. Still, the changes in ATP may develop differently in time and magnitude and additional factors may be released by PD, so our data do not necessarily inform what microglia do at synapses in the context of synaptic activity-driven ATP release under normal healthy conditions. Here, we directly show increased spine formation/elimination after laser light-induced prolonged microglial contact and for the first time demonstrate the role of microglia in the induction of bouton filopodia. The observed microglia-mediated increase in the rates of spine and bouton filopodia formation might provide a substrate for formation of new associations and may be mediated by BDNF released by microglia (*Parkhurst et al., 2013*). This hypothesis is based on the fact that BDNF release from microglia and TrkB phosphorylation in neurons can be strongly stimulated by the binding of ATP -

derived from the photodamaged cells (*Davalos et al., 2005*) to the purinergic receptor P2X4R (*Khakh and North, 2012*).

Interestingly, our results revealed that after Spine-PD there was higher microglia attraction toward the targeted spine in comparison with Near-Spine-PD and Near-Bouton-PD. Also, higher variability in responses was observed in the latter conditions, particularly after the Near-Bouton-PD. It could be explained by the ability of microglia to respond to neurotransmitters during acute injury (*Pocock and Kettenmann, 2007*). It is possible that after a single spine cut there is a release of neurotransmitters that stronger attracts microglia processes in comparison with Near-Bouton-PD and Near-Spine-PD. On the other hand, the microglial response after Near-Bouton-PD and Near-Spine-PD could vary due to a photodamage of astrocytes or various non-*Thy1*-EGFP-positive neurons.

Our observation of microglia-mediated spine remodeling is in line with recent indirect data on the role of microglia in eliminating synaptic components in the adult hippocampus and the observation that depleting microglia or inhibiting microglial phagocytosis prevented forgetting of remote memories (*Wang et al., 2020*). Acknowledging a difference in the mechanisms of focal injury and homeostatic microglia-synapse interactions, we still anticipate that the proposed methodology will be instrumental in further elucidating some aspects of synaptic surveillance by microglia (*Akiyoshi et al., 2018*; *Nimmerjahn et al., 2005*) and the mechanisms of microglia-independent and dependent forms of synaptic remodeling during synaptic injury. The latter may help to dissect the multiple contributions of microglia to memory formation and updating under normal and neurodegenerative conditions.

# Materials and methods

## Key resources table

| Reagent type (species) or resource | Designation | Source or reference | Identifiers | Additional information |
|---|---|---|---|---|
| Strain, strain background (Mice) | Gt(ROSA)26Sor$^{tm9(CAG-tdTomato)Hze}$ | The Jackson Laboratories | Stock No: 007905 RRID:IMSR_JAX:007905 | On C57BL/6 background |
| Strain, strain background (Mice) | Cx3cr1$^{tm2.1(cre/ERT2)Jung}$ | The Jackson Laboratories | Stock No: 020940 RRID:IMSR_JAX:020940 | On C57BL/6 background |
| Strain, strain background (Mice) | *Tg(Thy1-EGFP)MJrs* | The Jackson Laboratories | Stock No: 007788 RRID:IMSR_JAX:007788 | On C57BL/6 background |
| Antibody | anti-Iba1, rabbit polyclonal antibody | Wako | Cat# 019–19741, RRID:AB_839504 | IF(1:500) |
| Chemical compound, drug | PLX3397 | MedChemExpress | HY-16749/CS-4256 | |
| Software, algorithm | Fiji | http://imagej.net/Fiji | RRID:SCR_002285 | |
| Software, algorithm | R | (https://www.R-project.org/) R Foundation for Statistical Computing | RRID:SCR_001905 | |
| Software, algorithm | GraphPad Prism | GraphPad Software (http://www.graphpad.com/) | RRID:SCR_002798 | |

## Animals

Mice expressing enhanced green fluorescent protein (EGFP) under the control of a modified *Thy1* promoter region (*Feng et al., 2000*) (# 007788 from The Jackson Laboratory) and mice expressing red fluorescent protein dtTomato in microglia under the control of the endogenous *Cx3cr1* locus

(*Yona et al., 2013*) (obtained by crossbreeding of # 007905 and # 020940 lines from The Jackson Laboratory) were crossbred to simultaneously visualize microglia and neurons; 23 two-month-old male mice were used in this study. CreERT2-mediated recombination was induced by five consecutive i.p. injections of tamoxifen at P30. The mice were housed individually under a fixed 12 hr light/dark cycle with food and water available ad libitum.

In this initial study, we used males to maximize our chances to detect microglial effects, as it is known that microglia exhibit gender-specific responses to stimuli and male microglia have higher motility capacity (*Lenz and McCarthy, 2015*) and are more prone to inflammatory activation than female microglia (*Villa et al., 2018*).

## Tamoxifen preparation and administration for induction of CreERT2 activity

Tamoxifen (T5648, Sigma) was diluted in corn oil to make solution of 20 mg/ml, which was protected from light. Tamoxifen solution was freshly prepared the day prior to injections and placed on a rotator shaker to dissolve tamoxifen overnight at room temperature. Mice were then given intraperitoneal injections of 2 mg Tamoxifen (100 µl volume) per day, for a total of 5 consecutive days (*Madisen et al., 2010*).

## PLX3397 treatment

To study the role of microglia-neuron interactions, we used the drug PLX3397 (inhibitor of the receptor to CSF-1 of myeloid cells) (*Elmore et al., 2014*) from MedChemExpress (# HY-16749/CS-4256). Mice received PLX3397 mixed with Nutella (1:1000) via oral administration every day for 4 weeks (*Spiller et al., 2018*). The daily dose was between 1 and 1.5 mg of PLX3397 in 1–1.5 g of Nutella (~40–50 mg/kg of body weight) (*Mok et al., 2014*; *Spiller et al., 2018*). This mix and the control (Nutella) were added to polystyrene Petri dishes and placed on the cage floor. All mice received a dose of Nutella 48 and 24 hr before the start of the treatment, to avoid the novelty effect and initial avoidance. Our pilot study revealed that microglial cells that remained after PLX treatment were still attracted to the photodamage sites. To ensure that we completely deplete microglia in large areas of RSC in all mice used for synaptic remodeling analysis, we used the above-described treatment to control the exact amount of drug that each mouse consumed and performed longitudinal monitoring of microglia depletion in the RSC of treated mice every week.

## Surgery and in vivo two-photon imaging

Cranial windows were prepared in 2-month-old mice that were anesthetized with isoflurane at 1.5–2% and oxygen to 0.4 l/min. Mice were placed under a heating pad to keep the temperature of the mouse body at 37°C. The cranial window was located at 2 mm anteroposterior and 2 mm mediolateral from Bregma. After removing the overlaying skull, a round glass coverslip (5 mm diameter, Thermo Scientific) was fixed directly above the brain onto the skull by applying cyanoacrylate glue (Pattex-Henkel, Germany). Once the glue was dry, the surroundings of the coverslip were sealed with dental cement (Paladur, Heraeus Kulzer, Germany). The animals were then transferred to their cages and analgesics were given for three days (ketoprofen, 5 mg/kg of body weight). Then, all animals were allowed to recover for 1 month prior to imaging. Seven of the 23 mice used in this study, in which the craniotomy window was not optically clear at this time, were excluded and their brains were kept for immunofluorescence experiments (see the experimental design header).

In vivo two-photon imaging was performed using a multiphoton microscope (LSM 7 MP, Carl Zeiss, Germany) with a Ti:Sapphire laser (Chameleon Vision II, Coherent). Imaging of the retrosplenial cortex was performed with a 20x water immersion objective lens (Zeiss, N.A. = 1.0). The laser was tuned to 900 nm for EGFP imaging and to 850 nm for photodamage experiments (150–200 mW). dtTomato fluorophore was excited at 1040 nm. Thirty minutes before placing a mouse under the microscope, it was anesthetized intraperitoneally with ketamine (90 mg/kg body weight) and xylazine (18 mg/kg of body weight) in 0.9% NaCl solution. The eyes were protected with eye ointment (Bepanthen) and a heating pad (37°C) was placed under the mouse to maintain the body temperature. To avoid photodamage to the sample, the pixel dwell time was kept between 1 and 2.5 ms.

Imaging began 4 weeks after the craniotomy, which proved to be an optimal time for quantification of the turnover of dendritic spines and axonal branches (*Pryazhnikov et al., 2018*). In general, three fields of view containing several dendritic elements (second and higher order branches) and axonal branches per mouse were selected and recorded with low magnification (20x objective, zoom factor: 1.5; 1024 × 1024 pixels; image size 292.01 µm x 292.01 µm; a z-stack of 20 optical sections with 2 µm z-spacing). Care was taken to achieve close to identical fluorescence levels across imaged fields of view. For each field, two dendrites and one axon (separated by >250 um) were selected for further imaging. Then, high-magnification images of the previously selected dendrites and axon were collected in parallel using the position module of the Zeiss Zen software. Recordings were done before (baseline) and after photodamage during 2 hr every 10 min (20x objective, zoom factor: 5; 1024 × 1024 pixels; 85.02 µm x 85.02 µm; a z-stack of 5–10 optical sections with 1 µm z-spacing). Thus, recordings lasted 4 hr per animal, and two animals were imaged per day (*Figure 1A*). Typically, this protocol was repeated 2–3 times per animal (exact sample sizes are given in *Figure 1— source data 2*), then a new pair of mice was imaged.

## Laser ablation

In previous studies, laser ablation has been described as an effective method to dissect specific structures without altering the tissue adjacent to the impacted target (*Allegra Mascaro et al., 2010*). In this way, the cut of a dendritic branch (dendrotomy) (*Sacconi et al., 2005*) and axonal branches (axotomy) (*Allegra Mascaro et al., 2013*) have been reported. Here, spines were cut with a ≈ 1 µm-long line scan (laser power of 150–200 mW, excitation wavelength of 850 nm, 60 cycles, 1 s total duration of irradiation). For the Spine-PD experiment, a z-stack was acquired with the two-photon microscope to obtain a 3D reconstruction of the dendrite using a 20x water-immersion objective (Zeiss, N.A. = 1.0). Second, laser photodamage was performed by creating a line of ≈1 µm at the chosen segment. Near to spine PD and near to axonal bouton PD were performed using the same settings as for the spine ablation experiments in a neighboring field of view on the same day. For these experiments, the distance between the line scan and the spine or axonal bouton was ≈0.5 µm. To evaluate the success of our experiments, we measured the lifetime of each targeted structure. To accept a spine PD experiment as successful (75%), we considered that the spine should disappear during the first 30 min of recording. For the next to spine PD, the targeted spine should be stable during the first 30 min (≈80% of experiments). The average lifetime of all ablated spines was on average of 25.7 min in DP-SP experiments and 93.6 min for PD-NS experiments. In the case of the next to bouton PD, experiments that resulted in an evident axotomy (because of axonal swelling) were excluded (6%) and the lifetime of the targeted bouton was set to be more than 30 min for a successful experiment (total average = 118.62 min, 78% of cases). Special care was taken to introduce photodamage away from the dendritic branch and axonal shaft to avoid transient swelling.

Prior to the selection of spines/boutons for photodamage, we selected dendritic and axonal branches with visually similar spine/bouton density and shaft thickness. Only tertiary or secondary branches with no more than 2 µm of thickness and with the highest % of spines being located in one single optical plane (*Holmes and Berkowitz, 2014*) were chosen. Using these branches, the following inclusion criteria were taken into account in the selection of spines and boutons for photodamage: well-defined structures, not contacted by microglia (distance to microglia bodies from 23.9 to 75.8 µm) and without closely located neighbouring spines/boutons next to them to avoid their direct laser targeting. After the selection of the spines/boutons for photodamage, we imaged dendritic segments with the mean ± SD length of 27.34 ± 2.72 µm or axonal segments of 30 ± 2.98 µm. The dendritic thickness was in the range of 1.11–1.33 µm, on average of 1.18 µm. The spines and axonal boutons selected for PD were always located at the center of the segment studied (*Figure 2A2,B2, C2*) and were well-defined structures, with an average size perpendicular to the dendritic/axonal segment of 1.43 µm (1.10–1.82 µm) and 1.25 µm (1.03–1.47 µm), respectively.

To verify whether the structures selected for photodamage were within a spine/bouton clusters, the mean distance between the targeted structure and the closest left and right neighbours was compared with the mean of all the distances between neighbouring spines/boutons per each dendritic/axonal segment. The corresponding mean ± SD values and the Wilcoxon paired test p-values (with segments as sampling unit) were for Spine-PD: 1.88 ± 0.38 µm vs 1.53 ± 0.44 µm, p=0.009; for Near-Spine-PD: 2.07 ± 0.81 µm vs 1.77 ± 0.69 µm, p=0.023; for Near-Bouton-PD: 5.14 ± 4.39 µm vs 4.44 ± 3.83 µm, p=0.05. Thus, the structures selected for photodamage were not within clusters and

were even slightly more distant from their left and right neighbouring spines as compared to all spines/boutons in the selected segments.

## Experimental design for microglia depletion

To confirm the role of microglia in the synaptic remodeling related to the photodamage, 12 mice were selected and randomly subdivided into two groups of six animals each (simple randomization method), as follows: (i) Nutella mixed with PLX3397 and (ii) Nutella. The sample size was estimated based on previous studies (*Miyamoto et al., 2016*; *Wake et al., 2009*), in which 4–7 mice were used per group to find structural changes of spines after the contact with microglia or after the treatment with PLX3397. For this study, we needed a minimum total number of 12 animals to detect differences between the control and treated group with 80% power and a significance level alpha of 0.05 (G*Power version 3.0.10, the medium effect size of 0.25, F tests-ANOVA). Three days before the start of treatment, the pre-treatment measurements were performed: 2 hr of baseline followed by 2 hr of follow-up after the photodamage (3x Spine-PD, 3x Near-Spine-PD and 3x Near-Bouton-PD) using three fields of view per mouse (See Surgery and in vivo two-photon imaging header). In parallel, before and after the beginning of the treatment, one additional field of view (low magnification) was identified and tracked every 7 days for each mouse to follow-up the progression of microglia depletion. For this repetitive imaging, the positions were identified with the help of vascular landmarks and cell bodies of neurons. After 4 weeks of treatment, post-treatment baseline was recorded and laser photodamage (3x Spine-PD, 3x Near-Spine-PD and 3x Near-Bouton-PD) was performed again and followed by 2 hr of imaging. At the end of the post-treatment imaging sessions, all mice were euthanized and the brains were collected for immunofluorescence studies. To have enough time to obtain three biological replicates of each photodamage per mouse, mice were randomly separated in six subgroups and each subgroup was imaged consecutively (one control mouse and one PLX-treated mouse in each imaging session). Besides, four from the seven excluded animals (with not acceptable cranial windows, please see details in Surgery and in vivo two-photon imaging header) were used as a second control (Naïve group, *Figure 3—figure supplement 1*) for further immunofluorescence studies. These animals were fed with a normal chow diet and were euthanized at the same time as the control and PLX3397-treated mice.

## Image and data analysis

We collected for each mouse mostly three replicates of each type of photodamage and their respective baseline images before and after the treatment with PLX. The recording of each replicate lasted 2 hr (every 10 min) and produced image stacks of 5–10 Z-slices, 13 time points and two channels (EGFP and dtTomato). For image processing and parameter evaluation, we selected a region of interest (ROI) of 50 µm x 50 µm using these image stacks. A database was created to facilitate the annotation of findings.

## Normalized microglia-neuron area contact

To estimate the contact area between microglia and dendrites or axons, we quantified the colocalization area in µm$^2$ between the EGFP and dtTomato channels in each time point (13 values). For this purpose, a Fiji macro was used to set a colocalization threshold and run the Coloc2 plugin. We first manually removed neighboring dendrites and axons to isolate a single dendrite and axonal branch in the EGFP channel. Second, maximum intensity projection was generated from original z-stacks. Third, to normalize the contact area, we used the EGFP-positive area that represented the area of the studied dendrite/axonal segment. Finally, we calculated Microglia-Neuron interaction area in % by dividing the Microglia-Neuron interaction area by the area of the dendrite or axon and multiplying the result by 100. In addition, the mean Microglia-Neuron contact area in % was calculated by averaging areas estimated in 13 time frames (the sum of the contact area of each time point divided by the number of time points).

## Spine turnover

For each collection of stacks containing one ROI, images were aligned with each other using the dendritic branch and the Linear Stack Alignment with Sift and MultiStackReg Plugins (*Thévenaz et al., 1998*) in Fiji. In each ROI, we selected a dendritic segment of ≈30 µm of length.

Spines were manually counted using original *z*-stacks and the 3D convolution viewer in ZEN software (Carl Zeiss). Spine counting was performed without knowing the baseline configuration. To calculate spine turnover, lost and gained spines were identified and quantified using previously validated criteria (*Holtmaat et al., 2005*). Gained spines were considered as such when they had more than 0.4 μm (five pixels) (*Holtmaat et al., 2005*) in length and lost when the length was less than 0.4 μm (five pixels). The spine length was approximately calculated from the base at the dendritic shaft to the tip of the spine head. Besides, spines were scored as lost, if their position on the dendrite relative to neighboring spines shifted by ≥0.5 μm (*Cane et al., 2014*; *Holtmaat et al., 2005*). The percentage of spines appearing and disappearing from time point to time point were calculated as the turnover percentage (%). The turnover (%) was the sum of gained and lost spines from the previous time point to the analyzed time point divided by the sum of total spines from both time points and multiplied by 100 (*Fuhrmann et al., 2007*). Then individual percentages of gained (formation) and lost spines (elimination) were calculated by dividing the number of gained or lost spines by the number of total spines in each time point and multiplied by 100.

$$\text{Turnover}(\%)_{t1,t2} = \frac{(N_{\text{new}\,t2} + N_{\text{lost}\,t2})}{(N_{\text{total}\,t1} + N_{\text{total}\,t2})} \times 100$$

Also, the mean turnover rate (%) was calculated as the total sum of turnover rates (%) in each time point divided by the number of time points (13). In the same way, mean percentages of gained or lost spines were calculated by adding percentages of gained or lost spines in each time point and then dividing by the count of time points (13).

### Axonal bouton turnover

In this study, we selected segments of axons (~40 μm) that were clearly visualized, had *en passant* boutons (EPBs) with the area of 1–2 μm$^2$ and had no terminal boutons. To identify gained and lost axonal boutons across the time points, we used validated criteria (*Holtmaat et al., 2008*; *Holtmaat et al., 2006*), where gained boutons had to be three times brighter than the axonal backbone. To score a loss, brightness had to drop to below 1.3 times backbone brightness. However, to exclude the scoring of transport packets and moving organelles, EPBs included in the analysis had to be present on the same site for at least two consecutive time points (10 min). To distinguish between two close boutons, they have to be separated at least in one z-plane by 2 μm. The turnover (%) for axonal boutons was calculated in the same way as for the spine turnover. Percentages of gained and lost boutons were calculated as (boutons formed or boutons eliminated)/(total number of boutons observed across imaging sessions).

### Filopodia detection and measurements

Protrusions emerging from the spine head with a thickness of half or less than the spine head were considered as spine head filopodia (*Figure 1—figure supplement 1*). Only protrusions with the length of at least 0.5 μm were scored and measured. The same criteria were used for detection of filopodia formation from EPBs with the exception that the protrusions shorter than 1 μm were excluded (*Wu et al., 2012*). For analysis, three parameters were evaluated: filopodia occurrence, size of filopodia and filopodial stability. The filopodia occurrence (%) was calculated as the total number of replicates where at least one event (filopodia formation) appeared divided by the total number of replicates per mouse and multiplied by 100. The filopodia size in length was calculated from the border of the spine or bouton to the tip of the filopodia and expressed in μm for each timepoint. The maximal filopodia length value was used for statistical comparisons of filopodial size. Filopodial stability was expressed as a percentage, calculated by dividing the filopodial lifetime by the total time of evaluation, i.e. 120 min.

### Spine and boutons characterization according to stability

According to their stability in time, spines were classified into three types: stable, new and transient. Existing spines or boutons that lived for the two hours were classified as stable (100%). Newly formed spines or boutons across the two hours and after the first time point were classified as new. Spines living for less than 2 hr were classified as transient. To present a clear separation between categories, a cumulative stability percentage curve during the 2 hr was computed. For this purpose,

the stability of each type of spine or bouton was measured as the cumulative lifetime that each spine or bouton remained visible at each time point divided by the value of the timepoint, expressed in %. Finally, an average per replicate was computed, and then an average per mouse (*Figure 2A1,B1, C1*).

### Microglia-contacted spines and boutons

In order to calculate microglia-contacted spines or boutons percentage, a qualitative classification was performed. Microglia-contacted spines or boutons were identified when more than 20% of the spine or bouton area was in contact with microglia in at least one slice of the z-stack and in at least one time point (*Weinhard et al., 2018*). The contacted spine or bouton percentage (*Figure 2A4,B4, C4*) was calculated as the total number of spines or boutons contacted by microglia divided by the total number of spines or boutons and multiplied by 100.

In addition, microglia-contact duration (*Figure 2A5,B5,C5*) was evaluated as the total time in minutes that each individual spine and bouton was in contact with microglia. All parameters were averaged per mouse for further statistical comparisons (See Filopodia quantification).

### Immunofluorescence

Mice were intraperitoneally injected with ketamine (90 mg/kg body weight) and xylazine (18 mg/kg of body weight) in 0.9% NaCl solution and then perfused transcardially with PBS, followed by 4% paraformaldehyde (PFA). After perfusion, the brains were fixed in 4% PFA in PBS for 24 hr at 4˚C and then cryopreserved in 30% sucrose in PBS at 4˚C. Fixed brains were cut into coronal sections at 40 μm using a freezing microtome. Floating brain sections were stored in a cryoprotectant solution (30% ethylene glycol, 30% glycerol, 10% 0.2 M sodium phosphate buffer pH 7.4, in $dH_2O$) at 4˚C. The sections were washed three times with washing solution (0.1% Triton X-100 in 0.1 M Tris-buffered saline). Sections were incubated 30 min with a blocking 10% FCS solution. For immunofluorescence, sections were incubated overnight with the primary antibody (goat anti-rabbit Iba1, Wako, Cat. Number: 019–19741, dilution 1:500), washed and incubated 3 hr with Alexa Fluor 647-conjugated secondary goat anti-rabbit antibody (Life Technologies-Invitrogen, Cat.Number: A-21245, dilution 1:250). The sections were then washed and incubated with DAPI for 15 min. Slides were analyzed using confocal microscopy (Carl Zeiss LSM 700) using x20 objective (N.A. = 0.4).

### Statistical analysis

R 3.6.3 version and RStudio 1.2.5033 software were used for data management and statistical analysis. The Prism version 7.0 software (Graphpad, San Diego, CA) and package ggplot2 of R were used to create graphs. Here, all graphs and statistics are shown per animal as a unit of observation (the data from replicates from the same animal were averaged separately for Spine-PD, Near-Spine-PD and Near-Bouton-PD). Paired-samples two-sided Wilcoxon test was used for comparison of cumulative turnover (%), lost (%), gained (%) spines/boutons and filopodia occurrence (%) between baseline and after Spine-PD, Near-Spine-PD and Near-Bouton-PD. For comparison of temporal profiles of turnover (%) and neuron-microglia contact area (%) between different groups or experiments, mixed ANOVA with repeated measures and GEE (R packages: lmer4, geepack and emmeans) were used. Besides, GEE and linear mixed model analysis were performed using an independent within-group correlation structure to calculate the interaction between pre- and post-treatment values (time) and treatment groups (PLX and control) followed by a Tukey's post-hoc tests for pairwise comparisons (Model: time as repeated measures, treatment as a fixed factor, time as a covariate, and their interaction treatment*time). For all reported differences between groups, the p-values are <0.05.

## Acknowledgements

The project has been supported by the federal state Saxony-Anhalt and the European Structural and Investment Funds (ESF, 2014–2020), project number ZS/2016/08/80645 (to KDF and AD). The authors thank Weilun Sun and Dr. Janelle Pakan for comments on the manuscript.

# Additional information

## Funding

| Funder | Grant reference number | Author |
| --- | --- | --- |
| The federal state Saxony-Anhalt and the European Structural and Investment Funds | ZS/2016/08/80645 | Klaus-Dieter Fischer Alexander Dityatev |

The funders had no role in study design, data collection and interpretation, or the decision to submit the work for publication.

## Author contributions

Carla Cangalaya, Data curation, Software, Formal analysis, Investigation, Methodology, Writing - original draft; Stoyan Stoyanov, Conceptualization, Supervision, Investigation, Methodology, Writing - review and editing, SS established surgery and microscopy methods and performed pilot experiments; Klaus-Dieter Fischer, Supervision, Funding acquisition, Project administration, Writing - review and editing; Alexander Dityatev, Conceptualization, Resources, Supervision, Funding acquisition, Validation, Visualization, Methodology, Writing - original draft, Writing - review and editing

## Author ORCIDs

Alexander Dityatev (iD) https://orcid.org/0000-0002-0472-0553

## Ethics

Animal experimentation: All animals were treated in strict accordance with ethical animal research standards defined by the Directive 2010/63/EU, German law and approved by the Ethical Committee on Animal Health and Care of Saxony-Anhalt state, Germany (license number: 42502-2-1346).

## Decision letter and Author response

Decision letter https://doi.org/10.7554/eLife.58435.sa1
Author response https://doi.org/10.7554/eLife.58435.sa2

# Additional files

## Supplementary files

- Source code 1. R code for GEE analysis.
- Transparent reporting form

## Data availability

All measurements, statistical analyses and the R code generated and used in this study are included in the manuscript.

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
