## [Decision Letter]

**Acceptance summary:**

This is an interesting study with broad implications for the field of microglia synaptic interactions. The authors use a method of spatially limited synaptic or axonal photodamage and then characterize relation between microglial contact and neuronal remodeling using two photon in vivo imaging. Treatment of mice with the CSF1 receptor inhibitor, PLX3397 to eliminate microglia revealed that many neuronal changes are due to the response of microglia to the damage. Microglia respond rapidly to focal synaptic damage, form contacts with synaptic structures, and cause a focal remodeling of dendritic spines and axonal boutons around the injury site.

**Decision letter after peer review:**

Thank you for submitting your article "Light-controlled engagement of microglia to focally remodel synapses in the healthy adult brain" for consideration by *eLife*. Your article has been reviewed by three peer reviewers, including Beth Stevens as the Reviewing Editor and Reviewer #1, and the evaluation has been overseen by Gary Westbrook as the Senior Editor. The following individual involved in review of your submission has agreed to reveal their identity: Ania Majewska (Reviewer #2).

The reviewers have discussed the reviews with one another and the Reviewing Editor has drafted this decision to help you prepare a revised submission.

Summary:

This is an interesting study that will be of broad interest for the field of microglia -synaptic interactions. The authors use a clever method of spatially limited synaptic or axonal photodamage and then characterize relation between microglial contact and neuronal remodeling. This in itself would have been an interesting experiment, but the use of PLX3397 to eliminate microglia and then show that many neuronal changes are in fact due to the response of microglia to the damage, elevates this to a study that really provides insight into the roles of microglia in synaptic changes in this injury model. The authors present evidence that microglia respond rapidly to focal synaptic damage, form contacts with synaptic structures, and cause the very focal remodeling of dendritic spines and axonal boutons around the injury site.

Essential revisions:

Overall the study is rigorous, well performed and well described. However, there is general agreement that the discussion and title are misleading and need to be revised to make it clear that this is an injury model. These data do not necessarily inform what microglia do at synapses in the context of activity-driven ATP release under normal healthy conditions, but rather the authors use a pathological stimulus to attract microglia to the neuron which could result in very different types of remodeling than during normal plasticity. Moreover, there are several other points that need to be addressed and clarified including:

– Please provide rationale why male mice were used and address if there were any sex-specific differences observed. If not, please add more on this possibility in the Discussion and Materials and methods.

– Please clarify and provide more details in the Materials and methods as noted by reviewers (see specific comments below)

– Please make it clear in the main text that the resolution of light microscopy is insufficient to claim that colocalization of microglia and spines represent physical "contacts." Please use a different term.

– The distance plots in Figure 2(B,H,N) seem to be probability density plots. If so, why does the distribution look so strange for transient spines in 2N and is this normalized correctly?

– Please clarify how spines were selected for photodamage. Given the clustering of inputs reported in cortex, it would be informative and important to compare the observed values in Figure 2 to a null distribution based on shuffling spine locations for each spine type.

– Several references are either mis-attributed, referred to for the wrong reason, or missing.

– Please revise title and remove the word “healthy” brain

Please see below for more details on each of these points.

Reviewer #1:

The authors have developed a clever method to increase the frequency of microglia-neuron interactions at focal sites within the otherwise healthy adult brain to test important ideas in the field about the role of microglia in synaptic remodeling. By ablating microglia with PLX, they show that microglia are necessary for some aspects of photodamage-induced remodeling. These experiments show that microglia are involved in spine turnover in their photodamage paradigms; however, they need to be careful about extending their conclusions to claim that microglia might have a similar role (albeit less frequent) without any sort of damage as this remains untested. The impact of this work would be increased if the authors are able to demonstrate that microglia depletion affects spine turnover under natural conditions, but appreciate that this is beyond the scope of the present work.

1) Please make it clear in the main text that the resolution of light microscopy is insufficient to claim that colocalization of microglia and spines represent physical "contacts." Consider using a different term.

2) The distance plots in Figure 2(B,H,N) seem to be probability density plots. If so, why does the distribution look so strange for transient spines in 2N and is this normalized correctly?

3) Please clarify how spines were selected for photodamage. Given the clustering of inputs reported in cortex, it would be informative and important to compare the observed values in Figure 2 to a null distribution based on shuffling spine locations for each spine type.

Reviewer #2:

This is an interesting study with broad implications for the field of microglia synaptic interactions. The authors use a method of spatially limited synaptic or axonal photodamage and then characterize relation between microglial contact and neuronal remodeling. This in itself would have been an interesting experiment but the use of PLX3397 to eliminate microglia and then show that many neuronal changes are in fact due to the response of microglia to the damage, elevate this to a study that really provides insight into the roles of microglia in synaptic changes. The authors present evidence that microglia respond rapidly to focal synaptic damage, form contacts with synaptic structures, and cause the very focal remodeling of dendritic spines and axonal boutons around the injury site. Overall the study is rigorous, well performed and well described. I have only a couple of comments that need to be addressed.

1) It is mentioned that this work was done in exclusively male mice. As microglia have been shown to exhibit sex specific phenotypes in both development and adulthood, this seems to be a major oversight (Villa et al., 2018; Thion, Cell 2018). While it would be much better to include female mice in the study, I acknowledge that this may be difficult given the large sample size used by the authors (which is a really big plus!) and the difficult experimental design. At the very least the use of exclusively male mice should be better rationalized and discussed by the authors in the text and mentioned as a part of their Materials and methods.

2) While the writing in the text itself is nice and easy to follow – much of the Materials and methods are very obtuse. I could not figure out how the authors did their biological replicates with the three fields and three injury types, whether these were imaged concurrently or sequentially and why different animals were imaged on different days in six imaging sessions. This made it difficult to understand how the authors averaged their data into a per animal quantification (which was very much appreciated as many studies inflate their Ns). Could the authors explain this better or provide a schematic that makes it clearer? It is also very difficult to understand how the filopodia quantification was done. Likewise survival percentages are hard to understand as these don't seem to be done the same way as most survival graphs. It is also unclear whether there is any overlap between new and transient spines or whether these are completely separate categories. The criteria for what determines spines and their appearance and disappearance were also really great. But how were spines that may be overlaid in z determined? In Figure 1C1 there appears to be a new spine in the lower part of the dendrite that appears right next to something that was counted as a single spine but has two spine heads (maybe indicating two spines that were overlapped in z and then separated due to tissue distortion into two clear spines).The description of the microglia-neuorns contact % – it is unclear whether this is completed in 3D or 2D. % is also somewhat misleading as it is the perimeter of contact vs. the area of the structure (maybe normalized microglia-neuron contact would be a better name?).

3) The authors do a great job summarizing their findings at the end of the manuscript. In the Introduction, however, they convolve microglial homeostatic surveillance with responses to injury in a way that is deceptive in rationalizing their experiments. While the work does have relevance for normal spine remodeling, and many molecular mechanisms are shared in homoeostatic conditions and in the early injury response, the two are certainly not the same. The authors should make it clearer that their work does not necessarily inform what microglia do at synapses in the context of activity-driven ATP release under normal conditions (or any other mechanism that regulates microglia-neuron interactions), but rather they are using a pathological stimulus to attract microglia to the neuron which could result in very different types of remodeling than during normal plasticity. I don't think this diminishes the impact of the work but I think this should be a small but important rewrite. The introduction of normal plasticity is appropriate – it just needs to be put into the context of the study.

Reviewer #3:

This study uses laser induced injury to trigger physical interactions between microglial processes and dendritic spines or axonal boutons. The study is purely descriptive, in reporting increased plasticity of neuronal connections at sites of injury or very close by. This plasticity is arbitrarily attributed to increased microglial contacts with injury sites mainly due to reduction in observed post-injury plasticity following microglial depletion. Microglial processes rapidly respond and physically interact with sites of localized laser-induced injury and may directly or indirectly influence the rewiring and plasticity of nearby synaptic structures, but if and how this may occur is not shown here. It is therefore not suitable for publication in *eLife* in its current form.

Several phenotypic observations are being reported regarding differences in kinetics, contact areas, filopodia origins, shapes, numbers etc; what do all these phenotypic observations mean mechanistically is unclear.

Given that all this structural plasticity namely new and transient spines occurred so close to the PD, it is unclear whether this microglial contact is related to plasticity of the neuronal elements or the inflicted nearby damage itself. Furthermore, the videos show extensive physical disruption of the dendrite especially when the damage is delivered onto the spines; so the physiological relevance and suitability of this injury model and observations related to it for drawing conclusions about neuronal plasticity, synaptic surveillance, and remodeling in the context of anything other than tissue injury are rather questionable.

Several references are either mis-attributed, referred to for the wrong reason, or missing.

The figures are not all properly labeled or described. For example in Figure 1 C2 D2 E2 it is unclear which lines correspond to which of the two Y axis metrics.

Mouse surgery and imaging experiments sound technically appropriate.

[Editors' note: further revisions were suggested prior to acceptance, as described below.]

Thank you for submitting your revised article "Light-controlled engagement of microglia to focally remodel synapses in the adult brain" for consideration by *eLife*. Your article has been reviewed by one of the original peer reviewers, and the evaluation has been overseen by a Reviewing Editor and Gary Westbrook as the Senior Editor. The following individual involved in review of your submission has agreed to reveal their identity: Ania Majewska (Reviewer #2). The reviewers have discussed the reviews and the Reviewing Editor has drafted this decision to help you prepare a revised submission.

Revisions:

The authors have satisfied the main concerns raised by reviewers. In the revised manuscript, more clarity is needed on the issue of damage vs. homeostatic synaptic remodeling. As suggested by the comments of reviewer 2 below, please acknowledge (in the Introduction and the Discussion), that this injury model may use different mechanisms than homeostatic microglia-synapse interactions in the healthy brain.*Reviewer #2:*

The authors have satisfied the majority of my concerns. I believe there is still room to improve clarity on the issue of damage vs. homeoostatic synaptic remodeling. I suggest that the reviewers acknowledge that they are using an injury model that while subtle may use different mechanisms than purely homeostatic microglia-synapse interactions. This should be stated in the third paragraph of the Introduction. Also in the final Discussion paragraph it would be good to acknowledge this possible difference in mechanism.

I am still unclear on how the imaging was done. Each subtype of photodamage takes 4 hours to image. I assume the different types of damage are imaged on different days. Is the idea of the "imaging sessions" just that two animals are imaged over a weeklong period to get all the data. Then another set is imaged?

---

## [Author Response]

Reviewer #1:The authors have developed a clever method to increase the frequency of microglia-neuron interactions at focal sites within the otherwise healthy adult brain to test important ideas in the field about the role of microglia in synaptic remodeling. By ablating microglia with PLX, they show that microglia are necessary for some aspects of photodamage-induced remodeling. These experiments show that microglia are involved in spine turnover in their photodamage paradigms; however, they need to be careful about extending their conclusions to claim that microglia might have a similar role (albeit less frequent) without any sort of damage as this remains untested. The impact of this work would be increased if the authors are able to demonstrate that microglia depletion affects spine turnover under natural conditions, but appreciate that this is beyond the scope of the present work.1) Please make it clear in the main text that the resolution of light microscopy is insufficient to claim that colocalization of microglia and spines represent physical "contacts." Consider using a different term.

We now mention this limitation: “Despite the evident colocalization of microglia with spines or boutons in our experiments, the resolution of light microscopy used here does not allow us to claim that this colocalization represents physical contacts between these both cells.”

2) The distance plots in Figure 2(B,H,N) seem to be probability density plots. If so, why does the distribution look so strange for transient spines in 2N and is this normalized correctly?

Indeed, the plots shown are the estimated probability density plots. We appreciate your observation that one of the curves was “strange”, there was a technical problem of the software used. We corrected the y-axis maximum in Figure 2N and now the third curve is fully visible.

3) Please clarify how spines were selected for photodamage.

We added the criteria for the selection of spines and boutons for photodamage in the Materials and methods section:

“Prior to the selection of spines/boutons for photodamage, we selected dendritic and axonal branches with visually similar spine/bouton density and shaft thickness. Only tertiary or secondary branches with no more than 2 µm of thickness and with the highest % of spines being located in one single optical plane (Holmes and Berkowitz, 2014) were chosen. Using these branches, the following inclusion criteria were taken into account in the selection of spines and boutons for photodamage: well-defined structures, not contacted by microglia (distance to microglia bodies from 23.9 to 75.8 µm) and without closely located neighbouring spines/boutons next to them to avoid their direct laser targeting. After the selection of the spines/boutons for photodamage, we imaged dendritic segments with the mean± SD length of 27.34±2.72 µm or axonal segments of 30±2.98 µm. The dendritic thickness was in the range of 1.11-1.33 µm, on average of 1.18 µm. The spines and axonal boutons selected for PD were always located at the center of the segment studied (Figure 2B,H,N) and were well-defined structures, with an average size perpendicular to the dendritic/axonal segment of 1.43 µm (1.10-1.82 µm) and 1.25 µm (1.03-1.47 µm), respectively.”

Given the clustering of inputs reported in cortex, it would be informative and important to compare the observed values in Figure 2 to a null distribution based on shuffling spine locations for each spine type.

Due to the clustering of spines in the cortex (Gökçe, *eLife* 2016), we verified whether the structures selected for photodamage were within a spine/bouton clusters. As the number of some spine/bouton subtypes was limited, we avoided to use reshuffling and employed the following approach: The mean distance between the selected structure and the closest left and right neighbours was compared with the mean of all the distances between neighbouring spines/boutons per each dendritic/axonal segment. The corresponding mean± SD values and the Wilcoxon paired test P-values (with segments as sampling unit) were for Spine-PD: 1.88 ±0.38 µm vs 1.53±0.44 µm, p=0.009; for Near-Spine-PD: 2.07±0.81 µm vs 1.77±0.69 µm, p=0.023; for Near-Bouton-PD: 5.14±4.39 µm vs 4.44±3.83 µm, p=0.05. Thus, the structures selected for photodamage were not within clusters and were even slightly more distant from their left and right neighbouring spines as compared to all spines/boutons in the selected segments.

Reviewer #2:This is an interesting study with broad implications for the field of microglia synaptic interactions. The authors use a method of spatially limited synaptic or axonal photodamage and then characterize relation between microglial contact and neuronal remodeling. This in itself would have been an interesting experiment but the use of PLX3397 to eliminate microglia and then show that many neuronal changes are in fact due to the response of microglia to the damage, elevate this to a study that really provides insight into the roles of microglia in synaptic changes. The authors present evidence that microglia respond rapidly to focal synaptic damage, form contacts with synaptic structures, and cause the very focal remodeling of dendritic spines and axonal boutons around the injury site. Overall the study is rigorous, well performed and well described. I have only a couple of comments that need to be addressed.1) It is mentioned that this work was done in exclusively male mice. As microglia have been shown to exhibit sex specific phenotypes in both development and adulthood, this seems to be a major oversight (Villa et al., 2018; Thion, Cell 2018). While it would be much better to include female mice in the study, I acknowledge that this may be difficult given the large sample size used by the authors (which is a really big plus!) and the difficult experimental design. At the very least the use of exclusively male mice should be better rationalized and discussed by the authors in the text and mentioned as a part of their Materials and methods.

We greatly appreciate this comment. Indeed, microglia can exhibit different responses to stimuli according to gender, for that reason we preferred to use only one gender for all experiments. We chose males to maximize our chances to detect microglial effects in this initial study, as according to previous works the male microglia have higher motility capacity than female microglia (Lenz, Neuroscientist 2014) and male microglia are more prone to inflammatory activation than female microglia (Villa et al., 2018). We include this information in the Materials and methods and consider to study both genders in the follow-up studies on microglia-neuron interaction in neurodegenerative diseases

2) While the writing in the text itself is nice and easy to follow – much of the Materials and methods are very obtuse. I could not figure out how the authors did their biological replicates with the three fields and three injury types, whether these were imaged concurrently or sequentially and why different animals were imaged on different days in six imaging sessions. This made it difficult to understand how the authors averaged their data into a per animal quantification (which was very much appreciated as many studies inflate their Ns). Could the authors explain this better or provide a schematic that makes it clearer?

The photodamage experiments were performed sequentially. To have enough time to obtain three biological replicates of each photodamage subtype per mouse, mice were randomly separated into four subgroups and imaged in four imaging sessions (each subgroup was imaged in a single imaging session). We clarified these points in the Materials and methods.

It is also very difficult to understand how the filopodia quantification was done.

We introduced filopodia quantification in subsection “Filopodia detection and measurements” and added a scheme for clarification.

Likewise survival percentages are hard to understand as these don't seem to be done the same way as most survival graphs. It is also unclear whether there is any overlap between new and transient spines or whether these are completely separate categories.

The three categories of spines are separate categories. We changed the Y-axis title of the graph to Cumulative stability percentage to avoid misunderstandings. We clarified stability percentage quantification in the text:

“According to their stability in time, spines/boutons were classified into three types: stable, new and transient. Existing spines/boutons that lived for the two hours were classified as stable (100%). Newly formed spines or boutons across the two hours and after the first time point were classified as new. Spines living for less than 2 hours were classified as transient. To show a clear separation between categories, a cumulative stability percentage curve during the 2 hours was computed. For this purpose, the stability of each type of spine or bouton, was measured as the cumulative lifetime that each spine or bouton remained visible at each time point divided by the value of the timepoint, expressed in %. Finally, an average per replicate was computed, and then an average per mouse (Figure 2A,G,M).”

The criteria for what determines spines and their appearance and disappearance were also really great. But how were spines that may be overlaid in z determined? In Figure 1C1 there appears to be a new spine in the lower part of the dendrite that appears right next to something that was counted as a single spine but has two spine heads (maybe indicating two spines that were overlapped in z and then separated due to tissue distortion into two clear spines).

For spine counting, we manually identified the spines and boutons by scrolling through the z-stacks in chronological order. 3D convolution viewer in ZEN software (Zeiss) was used to help us to visualize spines in 3D stacks. In addition, Z-projections were generated in ImageJ and used to make a montage to visualize spine remodelling in 2D. This new information was added in Materials and methods. We confirmed that the aforementioned spines were two and we properly labelled them in the Figure 1C1.

The description of the microglia-neuorns contact % – it is unclear whether this is completed in 3D or 2D. % is also somewhat misleading as it is the perimeter of contact vs. the area of the structure (maybe normalized microglia-neuron contact would be a better name?).

Maximum intensity projections were created from original z-stacks to quantify areas of colocalization between both cells. To clarify this point, we added more details in subsection “Normalized Microglia-Neuron area contact” and we changed the variable name “microglia-neuron contact %” to Normalized contact area (%) according to the suggestion.

3) The authors do a great job summarizing their findings at the end of the manuscript. In the Introduction, however, they convolve microglial homeostatic surveillance with responses to injury in a way that is deceptive in rationalizing their experiments. While the work does have relevance for normal spine remodeling, and many molecular mechanisms are shared in homoeostatic conditions and in the early injury response, the two are certainly not the same. The authors should make it clearer that their work does not necessarily inform what microglia do at synapses in the context of activity-driven ATP release under normal conditions (or any other mechanism that regulates microglia-neuron interactions), but rather they are using a pathological stimulus to attract microglia to the neuron which could result in very different types of remodeling than during normal plasticity. I don't think this diminishes the impact of the work but I think this should be a small but important rewrite. The introduction of normal plasticity is appropriate – it just needs to be put into the context of the study.

We do appreciate this comment. Given that our findings are based on the use of very focal photodamage, our results are linked to an acute response of microglia processes to a focal synaptic injury.

We added references and information related to the acute response of microglia in the Introduction:

“Microglia are highly motile cells that regularly survey their environment within the central nervous system and are responsible for immune defence in the brain (Nimmerjahn et al., 2005). Activation of microglia in response to infection or during ischemia, stroke and CNS trauma results in release of different inflammatory signalling mediators (Davies et al., 2019; Greenhalgh et al., 2020) and can be associated with synaptic elimination by microglia (Brown and Neher, 2014; Wake et al., 2009b).”

We also modified the following lines in the Discussion:

“We anticipate that the proposed methodology will be instrumental in further elucidating the molecular mechanisms of synaptic surveillance by microglia (Akiyoshi et al., 2018; Nimmerjahn et al., 2005) and the mechanisms of microglia-independent and dependent forms of synaptic remodelling during synaptic injury. The latter may also help to dissect the multiple contributions of microglia to memory formation and updating under normal and neurodegenerative conditions.”

Reviewer #3:This study uses laser induced injury to trigger physical interactions between microglial processes and dendritic spines or axonal boutons. The study is purely descriptive, in reporting increased plasticity of neuronal connections at sites of injury or very close by. This plasticity is arbitrarily attributed to increased microglial contacts with injury sites mainly due to reduction in observed post-injury plasticity following microglial depletion. Microglial processes rapidly respond and physically interact with sites of localized laser-induced injury and may directly or indirectly influence the rewiring and plasticity of nearby synaptic structures, but if and how this may occur is not shown here. It is therefore not suitable for publication in eLife in its current form.Several phenotypic observations are being reported regarding differences in kinetics, contact areas, filopodia origins, shapes, numbers etc; what do all these phenotypic observations mean mechanistically is unclear.Given that all this structural plasticity namely new and transient spines occurred so close to the PD, it is unclear whether this microglial contact is related to plasticity of the neuronal elements or the inflicted nearby damage itself.

This is why we performed the experiments with microglia depletion to dissect these two factors. Indeed, some aspects of structural plasticity in our paradigm proved to be induced by the photodamage independently of microglia, however others proved to require the microglia.

Furthermore, the videos show extensive physical disruption of the dendrite especially when the damage is delivered onto the spines; so the physiological relevance and suitability of this injury model and observations related to it for drawing conclusions about neuronal plasticity, synaptic surveillance, and remodeling in the context of anything other than tissue injury are rather questionable.

We appreciate these comments and the related comments of reviewer 2 and more clearly discuss that our data represent a response to synaptic injury, which might share some mechanisms with a physiologically triggered response.

Several references are either mis-attributed, referred to for the wrong reason, or missing.

We corrected these mistakes.

The figures are not all properly labeled or described. For example in Figure 1 C2 D2 E2 it is unclear which lines correspond to which of the two Y axis metrics.

We modified the Figures 1C2, 1D2 and 1E2 according to these and reviewer 1 suggestions.

Mouse surgery and imaging experiments sound technically appropriate.

We appreciate this conclusion.

[Editors' note: further revisions were suggested prior to acceptance, as described below.]

Reviewer #2:The authors have satisfied the majority of my concerns. I believe there is still room to improve clarity on the issue of damage vs. homeoostatic synaptic remodeling. I suggest that the reviewers acknowledge that they are using an injury model that while subtle may use different mechanisms than purely homeostatic microglia-synapse interactions. This should be stated in the third paragraph of the Introduction. Also in the final Discussion paragraph it would be good to acknowledge this possible difference in mechanism.

We appreciate this comment and added the following sentences.

In the Introduction:

“Therefore, we developed an approach to evaluate the role of microglia in synaptic remodelling by transiently attracting microglial processes to synapses in a temporo-spatially controlled manner using a laser-induced focal injury. This is an acute injury model that while subtle is likely to involve a combination of injury-specific and more general mechanisms shared by homeostatic/synaptic activity-driven microglia-synapse interactions.”

In the Discussion:

“Acknowledging a difference in the mechanisms of focal injury and homeostatic microglia-synapse interactions, we still anticipate that the proposed methodology will be instrumental in further elucidating some aspects of synaptic surveillance by microglia (Akiyoshi et al., 2018; Nimmerjahn et al., 2005) and the mechanisms of microglia-independent and dependent forms of synaptic remodelling during synaptic injury.”

I am still unclear on how the imaging was done. Each subtype of photodamage takes 4 hours to image. I assume the different types of damage are imaged on different days. Is the idea of the "imaging sessions" just that two animals are imaged over a weeklong period to get all the data. Then another set is imaged?

The responses to three types of photodamage (PD) were imaged in parallel at three locations (separated by >250 um) using the position module of the Zeiss Zen software. To study the baseline and PD-responses in the same animals, we performed 2h-baseline and 2h-post-PD recordings, i.e. 4 hours per animal, in 2 animals per day (as shown now in Figure 1 A). Typically, this protocol was repeated 2-3 times per animal (exact sample sizes are given in Figure 1—source data 2), then a new pair of mice was imaged. This information is added in subsection “Surgery and in vivo two-photon imaging”.